# A scoping review of the levels, implementation strategies, enablers, and barriers to cervical, breast, and colorectal cancer screening among migrant populations in selected English-speaking high-income countries

Resham B. Khatri[1,2]*, Aklilu Endalamaw[1,3], Darsy Darssan[1], Yibeltal Assefa[1]

**1** School of Public Health, The University of Queensland, Brisbane, Australia, **2** Health Social Science and Development Research Institute, Kathmandu, Nepal, **3** College of Medicine and Health Sciences, Bahir Dar University, Bahir Dar, Ethiopia

\* rkchettri@gmail.com

## Abstract

### Background

Cancer remains one of the leading causes of mortality and morbidity worldwide with colorectal, cervical, and breast cancers accounting for significant proportion of preventable deaths. Early screening, diagnosis, and treatment could prevent many of these deaths. However, migrants face persistent disparities in the screening, early diagnosis, and treatment of these cancers. This study synthesizes evidence on cancer screening uptake, implementation strategies, as well as their enablers and barriers among migrants in English-speaking high-income countries (Australia, the USA, the UK, Canada, and New Zealand).

### Methods

We conducted a scoping review of studies published in any language between 1 January 2015 and 31 December 2024. Studies were retrieved from four databases: PubMed, Scopus, Embase, and Web of Science. Search terms were developed based on four domains: types of cancer (colorectal, cervical, and breast), migrant populations, screening coverage, and country of residence. The uptake of cancer screening among migrants in selected countries was determined. A thematic analysis was conducted to analyze the data and identify key themes related to the implementation of cancer screening strategies, as well as their enablers and barriers.

### Results

A total of 80 studies were included in the review. Migrants exhibited varied levels of utilization of cancer screening such as cervical cancer (41% − 84%), breast cancer

**Data availability statement:** This study does not contain any raw data. All relevant publicly available secondary data included in this paper are available in the Supporting information files.

**Funding:** The author(s) received no specific funding for this work.

**Competing interests:** The authors have declared that no competing interests exist.

(24%−87%), and colorectal cancer (4%−55%). Four themes related to the implementation of cancer screening strategies were identified: i) culturally tailored health education and communication, ii) trust-building initiatives with providers and health systems, iii) family and community support for acculturation and engagement, iv) awareness and knowledge on increased risk perception. Several barriers to the implementation of cancer screening strategies were identified, including lack of insurance, transportation challenges, difficulty in speaking and understanding English, inflexible work hours of health services, cultural taboos, stigma, poverty, and undocumented (illegal) status of migrants. Enablers of the implementation of cancer screening strategies included faith-based messaging on cancer screening, community partnerships, home-based fecal immunochemical test kits, availability of after-hours services, gender-concordant care, social networks, acculturation, and trust-building.

## Conclusions

The uptake of cancer screening (breast, cervical, colorectal) varied and had low among migrants (e.g., refugees, culturally and linguistically diverse populations). Targeted, culturally tailored approaches, expanding interpreter services, and fostering cross-sector collaborations (e.g., linking screenings to cultural events) are essential for addressing disparities in cancer screening among migrants. Culturally sensitive and adaptive, equity-focussed interventions on cancer screening should be prioritized by ensuring sustained funding, disaggregated data collection on the uptake of cancers screening and design and implementation of programs on targeting diverse population groups.

## Introduction

Cancer remain one of the leading causes of morbidity and mortality globally, and is emerging as a major health problem in low- and middle-income countries (LMICs) [1]. Migration from LMICs to high income countries (HICs), especially the United States of America (USA), the United Kingdom (UK), Australia, Canada, and New Zealand (hereafter collectively called English-speaking countries), is increasing, and migrants currently constitute one of the most important priority population groups [2]. For example, migrants constitute 30% of the population in Australia [3], 14% each in the and the USA [4] and UK [5], 27% in New Zealand [6], and 22% in Canada [7]. Despite the evidence that early screening for those cancers is effective in both prevention and treatment, disparities in screening uptake persist among many priority populations, including migrants [8].

Colorectal cancers (CRC), cervical cancers, and breast cancers are major types of cancers where screening programs are beneficial for early prevention and treatment. For example, in 2020, CRC was the third most common form of cancer worldwide [9], with global prevalence of 18 per 100,000 in 2022, with prevalence in UK (31), USA (27), Canada (29), Australia (35), and New Zealand (39) [10]. In

2022, cervical cancer had a global prevalence of 13 cases per 100,000, with higher rates observed in LMICs [11]. Early diagnosis via screening program such as fecal immunochemical tests (FIT) or colonoscopy for early diagnosis and effective treatment can reduce CRC related mortality by 40–60% [12]. Similarly, the cervical cancer claims 342,000 lives annually worldwide, while countries like the UK (8), USA (6), Canada (7), Australia (5), and New Zealand (5) reported lower rates (per 100,000) of cervical cancer deaths [11,13]. Cervical cancer is also preventable through human papilloma virus (HPV) vaccination and screening (pap test for HPV) [13]. Breast cancer is the most diagnosed cancer globally, exhibited varying prevalence: the UK (94), USA (96), Canada (89), Australia (101), and New Zealand (94) had higher rates than the global average of 46 per 100,000 [14] while mammography screening can reduce mortality by 20–30% [13,15].

Countries like Australia, the USA, the UK, New Zealand, and Canada have established screening programs for cancers. For instance, in Australia, key cancer screening programs include the National Breast Cancer Screening (BCS) Program, the National Cervical Screening Program, and Breast Screen Australia [16]. In the USA, the Colorectal Cancer Control Program and the National Breast and Cervical Cancer Early Detection Program aim to address cancer screening needs [17]. In the UK, the National Health Service (NHS) Bowel Cancer Screening [18], Cervical Cancer Screening Programme [19], and NHS Breast Screening programs are available through the NHS program [18,20]. In New Zealand, the National Screening Unit oversees breast and cervical cancer screening, and bowel cancer screening is being phased in nationwide [21]. In Canada, cancer screening programs are primarily managed at the provincial level, with initiatives such as the Breast Screening Program and varying CRC screening programs across provinces [22].

The uptake of cancer screening varies in these English-speaking countries. For instance, in 2022–2023, more than 1.9 million eligible people (50–74 years) were screened through BreastScreen Australia, representing 52% of the target population. Meanwhile, 73% of eligible participants aged 25–74 engaged in the NCSP, with 42% of the 6.3 million invited individuals taking part in the program [23]. In New Zealand, 69% of eligible women were screened for breast cancer [24], 74% for cervical cancer [25], and 58% for bowel cancer [26]. In 2023, more than three in four (75.8%) of women aged 21–65 were up-to-date with cervical cancer screening (CCS), 79.8% of women aged 50–74 had a mammogram in the past 2 years, and 72.6% of adults aged 50–75 received colorectal cancer screening (CRCS) according to the latest guideline [27]. For the fiscal year 2022/23, screening coverage was 66.6% for breast cancer, 67.0% for cervical cancer in those aged 25–49 and 74.9% for cervical cancer in those aged 50–64, and 72.0% for bowel cancer in those aged 60–74 [28]. In Canada, over one-third of eligible individuals miss timely CRCS and at least one-quarter of eligible women miss their recommended breast and CCS [29].

These English speaking high-income countries have made strides in population-level screening for cancers; however, several barriers still hinder screening service utilization among migrants and preventing many cancer deaths [30]. Improving cancer screening uptake among migrants is critical for achieving health equity among priority population groups, especially populations of migrant backgrounds, in these countries [31,32]. This study synthesizes the level of cancer screening among migrants, implementation strategies, and their enablers and barriers. The findings of this scoping review could strengthen cancer screening for early diagnosis among migrant populations.

## Materials and methods

We conducted a scoping review of available literature on cancer screening among migrants in selected English-speaking high-income countries. The review methods and reporting of the findings were guided by the Preferred Reporting Items for Systematic Reviews and Meta-Analyzes Extension for Scoping Reviews (PRISMA-ScR) checklist (Supplementary File, S1 Table) [33,34]. We adhered to the methodological framework of scoping reviews proposed by Arksey and O'Malley [35], which was further refined by Levac and colleagues [36].This framework includes six steps: (a) identifying research questions, (b) identifying relevant documents, (c) selecting documents, (d) extracting and charting data, (e) summarizing and reporting of findings, and (f) engaging public and expert involvement (optional).

### Identifying research question

The following question guided the scoping review: What are the levels of cancer screening coverage among migrants in selected English-speaking countries, implementation strategies for cancer screening, and their enablers and barriers?

### Identifying relevant documents

Search terms were organized under five domains: i) cancer types (e.g., bowel/colorectal, cervical, and breast), ii) migrants (e.g., migrants, culturally and linguistically diverse groups, refugees), iii) screening coverage (e.g., uptake, usage, utilization, acceptance), and iv) location (e.g., USA, UK, Canada, Australia, and New Zealand). The last author (YA), in coordination with a librarian, developed a search strategy, which was independently reviewed and verified by other authors. The search strategy was tested in PubMed and search terms were revised after initial searches revealed new terms. Studies were retrieved from four databases: PubMed, Scopus, Embase, and Web of Science. A detailed search strategy tailored for each database is provided in the Supplementary file (S2 Table). Searches were conducted across databases from 1 January 2015–31 December 2024. The first author (RBK) searched the records on 6 January 2025. Selection criteria were based on the following domains: population, interventions, outcomes, study design, geography (country of residence), publication year, and accessibility [S3 Table)]. The review included all types of primary studies irrespective of study design (e.g., qualitative, quantitative, and mixed methods) and language of publication. However, no studies were identified that were published other than English.

### Selection of documents

Selected records were imported into Covidence to screen the titles and abstracts for the identification of the list of studies. The first author (RBK) initially screened the titles and abstracts and identified the list of potentially eligible studies for the full-text review. The second author (AE) independently evaluated the eligibility of each title and abstract for the included studies. RBK and AE then independently assessed the full text of relevant studies using an Excel sheet approved by the author team. The author team then cross-checked for any discrepancies, discussed them, and reached an agreement through further review.

Citations, abstracts, and full-text articles were managed and reviewed using EndNote (X20) reference manager. No studies were excluded based on methodological quality.

### Extraction and charting of data

A template in a Microsoft Excel (Microsoft Corporation, 2020) sheet (covering first author, year, title, objective, design, study types, population, country, migrants, location, cancer types, sample size, people screened, time of screening, frequency of screening, healthcare provider, cost, facilitators/enablers, and barriers) was developed to extract data from each eligible study [S4 Table)]. After the full-text review, data from selected studies were extracted into the data extraction template.

### Summarizing and reporting the findings

Quantitative data were analyzed to identify the level of screening for cancers among migrants. Thematic analysis was conducted for qualitative data to generate themes related to implementation strategies for cancer screening, as well as their enablers and barriers. The thematic analysis involved specific activities such as familiarization with data, construction of themes based on a priori themes, indexing, charting, and interpretation [37]. First, descriptive statistics of included studies (e.g., design, countries, types of cancer, migrant groups) were presented in a figure. Second, the level of cancer screening uptake by type and migrant group in selected countries was presented. Third, themes related to implementation strategies were explained narratively, with their barriers and facilitators summarized in a table. Quantitative and qualitative findings were integrated and discussed in the discussion section.

## Patient and public involvement

This scoping review did not involve patients or the public or was unable to consult with experts working in the field to validate and confirm our findings. We have included this as a limitation of study.

## Results

Fig 1 presents the study selection process for this review (Fig 1). A total 80 studies were included in the review.

Descriptive analysis of the total studies included in the review is reported in the Supplementary file (S5 Table). Of total 80 studies, nearly half of them were conducted in the USA (47.5%), with no studies from New Zealand. The highest number of studies focused on migrants from African region (20%), followed by immigrants (country of origin unspecified) (16%), and South Asian migrants (14%). More than half of the studies (52.4%) focused on CCS, followed by breast cancer screening (BCS) (31.3%), and CRCS (16.3%).

## Cancer screening among migrants

Table 1 presents utilization rates of CCS, BCS, and CRCS in the USA, the UK, Australia, and Canada. Regarding of the CCS, Screening in the UK was up to 75% [38], in the USA it ranges from 41% to 82.7% [39–41], in Australia it ranges from 43.9% to 76.9% [42,43], while in Canada it ranges from (74–77%) [44,45]. This indicates that screening rates are heterogeneous among different migrant groups such as among Hindu women in the UK (75%) [38], Filipinas: 82.7%; Asian Indians: 66.8%, and Arab: 41%) in the USA [39–41], and Asian women in Australia (43.9%) [42], and migrants in Canada (74%) [44,45] [Table 1]. Additionally, BCS in the USA ranges from 24–35% to 86% [46–49], in Australia it ranges from 31.4% −54.5% [50], and in Canada it ranges from 35.3% and 87% [51,52], indicating that screening rates are heterogeneous among different migrant groups such as in the USA (Chinese: 85.5% [53], Arab women (lifetime):86% [46,47], recent screening:24–35%) [48,49]), Australia (Korean women clinical breast examination: 31.4% and biennial mammogram: 54.5% [50]) and Canada (Asian:87% [51], and Sub-Saharan Africans:35.3%) [52]) [Table 1]. Finally, CRCS in the USA ranges from 23% to 55% [54,55], and in Canada it ranges from 4% to16.6% [56,57], indicating that screening rates

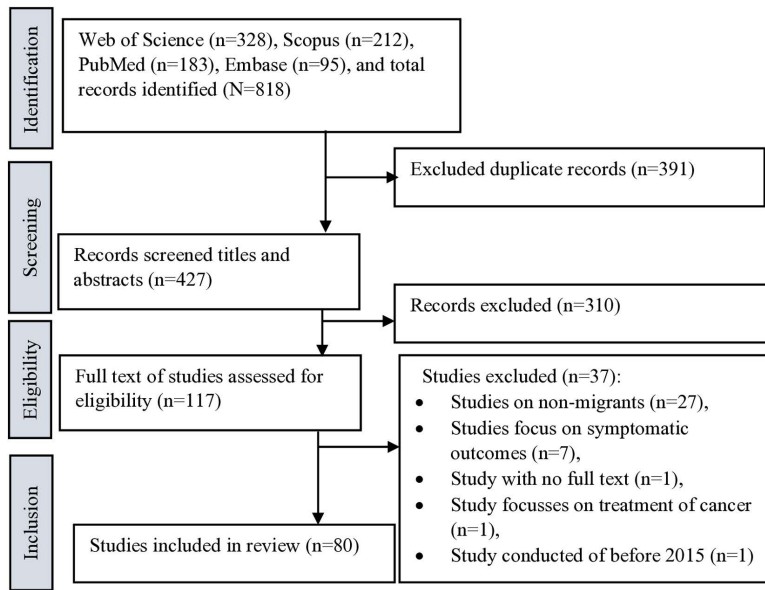

**Fig 1. PRISMA-ScR flowchart showing study selection.**

are heterogeneous among different migrant groups. For example, the screening rates in the USA are among Asian Indians (55%) [54], Chinese (50.9%) [53], and Vietnamese (23%) [55], and (59%) [58] while in Canada, the fecal occult blood test (FOBT) rate are is among refugees/immigrants (4%) [56] and Eastern Europeans (16.6%) [57].

## Summary of implementation strategies, and their enablers and barriers

Fig 2 summarizes the enablers and barriers to cancer screening in the UK, USA, Canada, and Australia [Fig 2]. Broader strategies for implementation included culturally tailored education and communication, trust-building initiatives with providers and the health system, family and community support for acculturation and engagement, and awareness and knowledge of increased risk perception.

## Culturally tailored health education and communication

Culturally tailored health education has improved cancer screening rates and awareness among diverse populations. In the USA, tailored health education approaches were found to improve uptake of mammography among Chinese immigrants [53,70], while CCS improved among Karen-Burmese, Bhutanese, and Somali refugees by addressing cultural and religious barriers [71–73]. In Australia, culturally sensitive programs enhanced engagement among CALD women [65,74–76]. Culturally relevant education interventions were effective in addressing barriers and boosting cancer screening rates for South Asian immigrants [42,47,77] and Muslim women in Canada [52,78,79]. CCS interventions such as HPV self-sampling and peer-led initiatives built trust and overcame changing socio-cultural norms and stigma against cancer screening among Turkish, Arab, and Somali women [40,47,80], refugees [72,81], Spanish-speaking women [73],

**Table 1. Percentage of uptake of cervical, breast and colorectal cancer screening among migrants in selected English-Speaking high-income countries.**

| Country | CCS | BCS | CRCS |
|---|---|---|---|
| UK | • 75% Hindu women [38] | | |
| USA | • Pap test use:<br>  Asian (71.7%),<br>  Filipinas (82.7%),<br>  Asian Indian (66.8%),<br>  Chinese (68.7%), and<br>  Other Asian (68.1%) [39,40].<br>• East African: 63% [59], 83.8% in U.S./SSA [39,60]<br>• Latinos (English/Spanish-speaking): 51% [41]<br>• MENA- Turkish (66.1%) [39,40]<br>• Arab Refugees: 41% [41]<br>• Bhutanese refugees: 44.4% [61]<br>• Special Immigrants/ Refugees:60% [62]<br>• Other migrants:<br>  -White (83.2%), AI/AN (81.2%) [39,40].<br>  -Black (84.3%) [39,60]. | • Chinese: California (85.5%) [53].<br>• South Asian(48.5%) [63]<br>• Caribbean/Latinos (63.7%) [63,64].<br>• Dominican Latinos: (76.5% (past year) [63,64].<br>• MENA: Muslim Arab lifetime (86.4%) [46,47]<br>• Refugees: Arabic-Speaking lifetime (86.6%) [46,47].<br>• CALD: Lifetime (81%)<br>  Annual (24%)<br>  Biannual (35%) [48,49] | • Asians in Chicago: 30% (vs. 59% general) [58].<br>• Asian Indian colonoscopy: 55%) [54].<br>• Chinese in California: 50.9%<br>  Men 54.2%<br>  Women: 47.9%) [53].<br>• Vietnamese: Colonoscopy 23%,<br>  FIT 6%, screening 22% [55]. |
| Australia | • Asian:43.9% [42].<br>• Refugee: 61.8% [65]<br>• CALD:76.9% (4–17 year gaps) [43].<br>• Non-refugee:73.6% [65] | • Korean Australians:<br>  Annual clinical exams:31.4%<br>  Biennial mammograms: 54.5% [50].<br>• African: 65.9% [66].<br>• CALD: 60.6–69% biannual [67,68]. | |
| Canada | • CALD migrants/refugees:74.3% [44], 77% [45]. | • Asian<br>  East Asia & Pacific:56.0% [52].<br>  Immigrants: 87% [51]<br>• SSA: 35.3% [52].<br>• Refugees (mammogram): 59% [56] | • East Asia & Pacific: 25.5% [57]<br>• Refugees/immigrants: 4% FOBT [56]<br>• CALD:22% up to date on FOBT [69].<br>• Eastern Europe & Central Asia: 16.6% [57] |

**Fig 2. Summary of enablers, and barriers of cancer screening among migrants in English speaking high-income countries.**

Korean Americans [48], Iraqi/Syrian refugees [82], Arabic/Italian women [83], and Latina women [64]. Similar successes were seen in the UK and Canada, emphasizing community health workers and tailored strategies for cancer screening to address myths and build trust [84,85]. Culturally tailored education for cancer screening such as peer-led programs and community health workers was found effective raising awareness against stigma on cancer screening among migrants such as Turkish women [40], Arab women [47,80], Afghan refugees [86], countering fatalism and stigma in the USA [87],

and overcoming cultural taboos (e.g., taking Pap smears and mammograms by male providers, and concerns of privacy) and reliance on family for health decisions in UK [88]. Such culturally tailored educational interventions influenced and motivated participation in screening for cancer among Pacific Islander women [89], leveraged for decision making among migrants of Latino [90], and Muslim backgrounds [91].

Furthermore, personalized and accessible communication has proven effective in increasing cancer screening rates. For example, it led to higher mammogram booking rates in Australia [83], and was effective in increasing CCS among women from Chamorro/ Samoan/Tongan [89] and South Asian backgrounds [42]. Addressing knowledge gaps, particularly among individuals with lower-level education and limited English proficiency, has been critical [42,65,75,76]. Tailored communication was reported to increase awareness and self-monitoring of disease conditions, thereby helping to reduce language barriers [49,58,83,92]. These strategies were effective in addressing challenges in communication and understanding English among Korean Americans [92]), Iraqi/Syrian refugees [82], Karen-Burmese refugees [71], Serbian/ Macedonian communities [93], Bhutanese refugees [61,94], and Haitian women [95]. Culturally tailored communication through peer groups and community members was found to increase health literacy and encourage cancer screening (e.g., mammograms) among Spanish-speaking migrants [73,96], Thai women in Australia [97], and Chinese migrants [98]. The provision of interpretation services has been successful in addressing linguistic and communication barriers, tackling on misinformation, thereby improving health access to enhance CRCS rates for Arab women [68,83], Haitian Americans [95], Arab migrants [57,77], and raising awareness among ethnic minority groups [84,88,93].

Nevertheless, the lack of culturally tailored education contributed to increased equity gaps migrant populations often avoiding screening [46,47], among new/recent immigrants [99], South Asians [63], South Asians in the UK [38], and also among Somali and Latino migrants in the USA [72,100]. The lack of culturally focused screening interventions led to negative attitudes toward checkups, which influenced delays in screening (e.g., mammography) among Bhutanese refugees in Australia [94,101], women from the MENA region in Canada [80] and among Myanmar Americans [102]. Such culturally insensitive information was reported to delay the uptake of screening among immigrant Arab women facing reduced mammography use [77], Polish women in Scotland [103], and South Asian American Muslim women [87], and Black and South Asian women using colonoscopy services [88]. Modesty concerns regarding healthcare, along with reduced cancer screening rates, were noted among Muslim women in Canada and the USA [52,79], promiscuity concerns among Latinas [90], South Asian Muslims and Arabic-speaking refugees [41,87], and among Somali immigrants in the USA [81,100]. Poorly personalized language and communication barriers contribute to low screening coverage among South Asian faith groups in the UK [84], Serbian/Macedonian communities in Australia [93], and immigrants and refugees [56], and Thai migrants [97], perceived obstacles among Arabic-speaking refugees [46]. Inadeqaute culturally health education and awareness program were reported to contribute to low screening rates and gaps in uptake persists among Polish women in Scotland [103], and immigrants and refugees in Canada [44,56]. Delayed or reduced care reported in screening programs [68,104], South Asian migrants [84], Serbian/Macedonian [93], Asians [39], and refugees from Bhutanese backgrounds [94].

## Trust building initiatives with providers and health system

Trusted health care providers, networks of care workers, and health systems have played a crucial role in overcoming cultural taboos on cancer screening, thereby increasing access to health services [45,52,53,72,78,79,104–107]. Greater trust in healthcare networks plays a vital role in compliance with cancer screening and HPV testing for CCS among East African immigrants [40,59,65,69,72,75,76,86,106]. Additionally, providers' cultural competence and communication skills were found to increase trust in the health system and engage Chinese American women [108] and Black women from sub-Saharan Africa in preventive health practices [60]. The provision of insurance and addressing structural inequities in CCS were important for refugees and Black African immigrant women in Canada and Australia [45,94,109]. Improving primary care access was essential for addressing barriers, such as work schedules, for South Asian migrants in the UK and

the USA [39,66,110]. Furthermore, trust in care providers was found to be effective in addressing privacy concerns and encouraging the utilization of BCS by CALD women [103,107,111]. Migrants who have trust with the health system were found to seek more care for cancer screening among Asian and African migrants in the USA [61,112]. Trust in care providers and primary care access were associated with higher uptake of cancer screenings and adherence among migrants from Muslim backgrounds in Canada [52,78], among Black African immigrants and older immigrant women in the USA [59], and migrants in Australia [62].

Poor trust with health care providers and the health system has led to reduced participation in cancer screening [52] among immigrant and refugees women in Canada [56,75,99]. East African men in the USA also exhibit low screening rates due to distrust in healthcare systems [106].This distrust is further evident in systemic barriers that hinder access to and engagement with preventive healthcare services among Korean immigrants [113] and refugees of Myanmar backgrounds [102].

## Family and community support for acculturation and engagement

Social support, family influence, and acculturation have significantly improved cancer screening. These factors helped normalize screening and boost adherence among British-Pakistani women [105], increasing BCS in the USA [48], and improving screening rates among Somali migrants and Latino immigrants [45,90,112]. Supportive environments for CCS were created for Bhutanese refugee women in Australia [42,74], where acculturation, family encouragement in healthcare, and social capital further enhanced participation in cancer screening [62,76,86,112]. Family and community engagement in screening programs have improved cancer screening participation among diverse populations, including Karen-Burmese and Bhutanese refugees [71], and among migrants in Australia [44]. In the USA, these programs leveraged family support to build trust and encourage screening among Somali refugee women [72] and Latinos [73,90,96]. Peer-worker programs have empowered Somali communities in Canada by increasing knowledge and strengthening awareness [56,63,72,85]. Similar societal and community interventions have improved participation in CCS and BCS among women from British Pakistani backgrounds [105], Myanmar American [102], and Bhutanese refugees backgrounds [62,114]. These programs have enhanced awareness, integrated preventive care, and boosted screening uptake [41,115].

Despite some positive outcomes, poor social support and acculturation contribute to significant equity gaps and reduced adherence to CCS among East African immigrants [59], and inadequate up-to-date of CCS among refugees [62]. Low screening rates were prevalent among migrants [46,47], with Black women reporting distrust in healthcare systems and concerns about costs [60,90,95,96,113]. Lack of family and community engagement have resulted in lower screening rates (such as FOBT and mammography) and higher non-adherence among migrants from African backgrounds in Australia [63,66], Muslims and refugees in Canada [57,69]. Newer immigrants and those from Muslim-majority countries have faced similar barriers that influenced lower uptake of cancer screening [65,78], such as South Asian and Latino migrants in the UK [38,116], and Somali refugee women [72]. Furthermore, the lack of effective family and societal interventions has led to struggles in expressing health needs and limited access to health services, resulting in low screening rates among immigrants from China in the USA [108], from Africa in Australia [107,110], and CALD women in Australia [68].

## Awareness and knowledge on increased risk perception

Increasing awareness and knowledge to enhance risk perception has been linked to higher cancer screening uptake [47]. For instance, efforts to raise awareness about cancer risk perception have demonstrated positive attitudes toward health check-ups and led to greater participation in cancer screening among Korean migrants [50] and African Australians [66]. Similarly, improved awareness and health literacy have been found effective in reducing stigma, increasing risk perception, countering misinformation, and enhancing understanding and healthcare access, all of which have contributed to increased cancer screening rates [42,65,70,75,76,114,117], especially among refugees [82,116]. The implementation of Health Belief Model interventions has been instrumental in bridging knowledge gaps, improving risk perception and susceptibility, and boosting confidence [48,67,92].

However, a lack of awareness and knowledge about risk perception has contributed to lower BCS rates and barriers to screening among women who are recent immigrants [99], Muslim and South Asian in the USA [72]. Despite awareness, low participation persists among Asian immigrants due to constraints such as time limitations and information gaps [42,76,110,116,117]. Similarly, immigrants from Somali, Black African, and Bhutanese refugee backgrounds in the USA often delay screening until symptoms appear [94,100,115]. Chinese Americans showed reduced screening adherence [53], while Haitian women in the USA had low compliance with referrals [95], and South Asian faith groups in the UK [84]. Similar barriers hindered cancer screening participation among South Asian faith groups in the UK, with South Asian immigrants facing delays in screening until symptoms appear [42,111,117]. These barriers also affect migrants, such as Bhutanese refugees [94], African immigrants [107], and women from MENA region [80]. Migrants from East Africa and the MENA region in Australia often rely on symptomatic care [82], while Asian migrants from Vietnam and Bhutan show lower screening rates [55,101].

Table 2 details the list of enablers and barriers of cancer screening under each broader theme of implementation strategies [Table 2]. Enablers of the implementation of cancer screening include culturally tailored education, and multilingual communication (e.g., co-designed programs, mHealth tools, faith-based messaging, translated reminders, phone interventions) and trusted provider relationships (e.g., female clinicians, interpreters, flexible appointments), all of which improve participation. Family and community support (e.g., peer networks, acculturation) enhance engagement. Awareness campaigns through social media and community education boost risk perception, while self-sampling options increase accessibility. On the other hand, barriers include cultural stigma (e.g., modesty, fatalism), distrust in healthcare, and language gaps that hinder screening. Systemic issues such as cost, time constraints, and inefficient healthcare systems, along with low health literacy and a lack of targeted outreach for immigrants, further reduce screening uptake. Addressing these challenges requires culturally sensitive, community-led strategies and policy reforms.

## Discussion

This review identified disparities in the uptake of cancer screening among migrant populations in English-speaking HICs. Stark disparities highlight ethnic, migration status, and geographic variations among migrants in these countries. The current review identified that low CRS rates exist for migrant women in the USA (41%) [39–41], Australia (43.9%) [42,43], and also in Canada (74%) [44,45]. Additionally, for BCS, the USA ranges from 24 to 35% [46–49], Australia ranges from 31.4% to 54.5% [50], and Canada had 35.3% [51,52]. For CRCS, the USA ranges has as low as 23% [54,55], and Canada ranges from 4% to16.6% [56,57] which indicate that screening rates are heterogeneous among different migrant groups. In comparison, there was high coverage of the breast, cervical and CRC screening in the USA, UK, Australia and Canada. For instance, for CCS, Australia had 73% [118], the USA had 73% [119], Canada had 87% [120], and the UK had 74.9% [121]. Similarly, for BCS,there was 50% participation in Australia [122], 80% in the USA [119], 69.9% in the UK [121], and 64% in Canada [120]. For CRCS, the national level coverage was 60% in the USA [119], 40% in Australia [123], 71.8% in the UK [121], 53.7% in Canada [120]. Additionally, migrants, refugees, non-English-speaking individuals, and those recently arrived have low participation rates in breast, cervical, and colorectal cancer screening compared to the general populations. Low screening rates were attributed to factors like lack of awareness, cultural stigma, invasive procedures, and systemic barriers including insurance gaps, language difficulties, and discouraging cultural beliefs. This stark equity gaps of the uptake of screening services among migrants and general populations suggests need for focussed service interventions towards universal access to screening services.

Culturally tailored health education, and multilingual communication interventions can address language and communication barriers for improved cancer screening uptake among migrants by ensuring clarity, accessibility, and cultural relevance by integrating community-specific values, trusted messengers, and accessible formats. Cultural stigma, particularly around CCS due to modesty concerns, is prevalent among Middle Eastern and South Asian women in English speaking countries [41,117]. Culturally tailored interventions, such as faith-based messaging (e.g., mosque-delivered

**Table 2. Implementation strategies and their enablers and barriers of cancer screening in the USA, UK, Canada, and Australia.**

| Strategies | Enablers | Barriers |
|---|---|---|
| Culturally tailored education and communication | • Culturally targeted education programs, tailored in-person, and online forums [40,47,53,70,74,80,88,98], and culturally sensitive co-designed community interventions, with community stakeholders and health workers [65,74–76,83,109].<br>• Culturally sensitive communication tailored to specific cultural and religious contexts (e.g., faith-based messaging, culturally tailored narrative adapted videos, mobile apps, and community health educators) [71–73,77,81,84,91,92].<br>• Mobile phone multilevel, multimedia messaging using mhealth tools (e.g., mobile apps and translated reminder letters) and phone booking [48,79,82,83,92].<br>• Verbal, interactive communication within community settings, follow-ups for non-compliant subgroups, culturally sensitive faith-centered communication, and religiously tailored family-inclusive and interactive approaches delivered [42,72,74,79,81,84,86,87,91].<br>• Mobile phone interventions in preferred languages to navigate health systems [49,83,92].<br>• Tailored follow-ups for non-compliant subgroups [42,89], accessible information (e.g., social media ads) and community-based education [42,65,75,76]<br>• Co-designed translated reminders, phone calls, and flexible delivery (e.g., online forums), translated materials and workshops [68,77,83,93], through mobile apps mhealth tools [82,92], and culturally adapted videos [71], deploy interpreters [61,94], and simplify referrals processes and low-literacy educational resources [95].<br>• Education using narratives and engaging male partners to build trust [73,96], services tailored to their socio-cultural contexts, language-specific support [97].<br>• Interactive, linguistically appropriate programs, addressing language barriers and cultural norms [98,111].<br>• Culturally sensitive materials and workshops [77], multilingual guides for primary and primary care [57,76].<br>• Simplified CRCS kit instructions and promotion in ethnic media (local language broadcasts) [84,88,93]. | • Cultural stigma, religious, cultural and faith-based beliefs, and social stigma [41,46,47,72,77,81,100], lack of awareness, limited access to female providers, language gaps [63], lack of culturally sensitive programs, and culturally appropriate providers [52,79].<br>• Lack of targeted outreach to newer immigrant populations [99], HPV-related stigma on relationships [38].<br>• Beliefs linking pap smears to promiscuity, and cultural norms around modesty and religious fatalism [72,100].<br>• Cultural beliefs about the sacredness of the body, cultural misunderstandings [94,103],<br>• Cultural fatalistic views and negative attitudes toward checkups [102], cultural norms and low awareness [101]<br>• Lack of religiously tailored interventions and modesty concerns [87,90], reliance on family for health decisions [88].<br>• Language barriers and low awareness [84], poorly translated instructions and labelling confusion [93].<br>• Time constraints, language barriers, and poor communication [56].<br>• Unfamiliarity with screening practices and language barriers [97], perceived obstacles persist due to language gaps among Arabic-Speaking Refugees [46].<br>• Lack of translated materials creates screening gaps [103].<br>• Poor communication, language differences, and cultural misunderstandings [56], non-English-speaking households [44], lack of familiarity with preventive care, linguistic barriers, and cultural norms [68,104].<br>• Reliance on family for translation [84], poor-quality materials [93], reduces access [39], and trauma delays care [94] |
| Trust building initiatives with providers and health system | • Familiarity with screening guidelines, awareness and self-monitoring [47,58,77], home based self-sampling for accessibility and convenience, self-reports [38,58,112].<br>• Trusted health care providers access to female providers and culturally appropriate care [72,79,107],gender-concordant care, ethnic- and gender-tailored community programs sharing information by GPs, family, and mosques [52,64,78,89,105].<br>• Flexible appointments, opportunistic screening, provider recommendations, cultural competence and communication skills, provider-initiated primary care model [52,60,66,78,86,108,114].<br>• Dedicated clinics and early resettlement interventions (e.g., insurance access), combating systemic racism (e.g., unemployment) [39,45,76,94,106,110].<br>• Partnerships with community organizations-building trust and engagement, social connectedness, community networks [69,70,75,106,115], building trust with providers to improve cultural competence in clinical encounters [59,111].<br>• Convenience and familiarity of health system, longer engagement and access to specialist services, and prior screening history [59,61,62,103,112], regular physician visits, enrolment and establish refugee clinics for primary care, transportation, multilingual resources [40,45,52,53,65,72,75,76,78,104]. | • Fee-for-service healthcare systems [52].<br>• Lack of female providers (male/internationally trained physicians) and lack of primary care enrollment [75,99].<br>• Time constraints [56], distrust in the system and fear losing control and feel vulnerable in healthcare settings [108] and lack of insurance [106,113].<br>• Lack of targeted outreach to newer immigrant populations, lack of insurance for recurring screenings [46,47,99].<br>• Distrust in healthcare systems and concerns about cost [60,90,96,102,113].<br>• Inefficient healthcare systems, inefficient healthcare software [84,111]. |

*(Continued)*

**Table 2.** (Continued)

| Strategies | Enablers | Barriers |
|---|---|---|
| Family and community support for acculturation and engagement | • Family networks and mosque communities for raising awareness [48,105].<br>• Social support from family, friends, and community networks, face-to-face communication and engagement [45,84,88,90,112], group-based models and community-based events [42,74].<br>• Longer residency in the host country and greater acculturation [62,112], and socially stable refugees (married, long-term residents) [62,86].<br>• Educational intervention fostering community-wide support [89].<br>• Family support for motivation and respecting women's autonomy in decision-making narratives [73,85,90].<br>• Provider-led culturally adapted videos during early resettlement [71].<br>• Sustained public health campaigns [44], multilevel interventions involving families and community leaders [71,72].<br>• Engage male partners to build trust and family encourage screening [73,86,96].<br>• Involve families and leaders using peer workers [72,85], cross-sector collaboration, community engagement, and peer-worker programs [56,63].<br>• Targeted education campaigns, close-knit networks for tailored interventions, culturally specific insights to understand unique cultural perspectives [41,62,102,115]. | • Persistent gaps in care continuity and low adherence [59,62].<br>• Language, lack of insurance, trauma or cultural stigma around recurring screenings [46,47], systemic racism based language [60,90,96,113].<br>• Low-income neighborhoods' employment status, financial constraints [66,75,95,99].<br>• Refugee status, recent immigration, and no regular physical exams [63], experience systemic exclusion systemic inequities [57,65,69,78].<br>• Fear the pain associated with screening, discomfort and past trauma [38,116].<br>• Fear and embarrassment due to cultural norms and concerns about privacy and lack of preferred gender preferences for providers [68,72,107].<br>• Work-life balance, and language barriers persist even among fluent speakers [110]. |
| Awareness and knowledge on increased risk perception | • Increased knowledge about breast cancer signs and mammograms [47], higher knowledge and positive attitudes toward health check-ups [50], education and awareness for fostering positive attitudes [66].<br>• Awareness raising through social media and community-based education [65,75,76,114].<br>• Targeted educational interventions for improved risk perception, and age-specific information cancer screening [42,70,116,117].<br>• mhealth solutions and mobile apps and interventions targeted on breast awareness self-breast examination education [48,67,82,92]. | • Misconceptions about sexual inactivity and lack of reminders [110,116], lack knowledge about screening, low awareness of screening benefits, lack of awareness on preventive services [55,82,84,100,101,111,115].<br>• Limited knowledge of screening practices and diseases, low-risk perceptions, lack knowledge and constraints (e.g., time, information gaps) [42,76,80,95,117]<br>• Low health literacy, cultural beliefs (e.g., body sacredness), and low awareness of screening protocols [53,80,94,107] |

education in Muslim communities) and narrative videos addressing cultural myths, have been shown to increase screening [124]. Cultural relevance has been a key factor, with faith leaders endorsing screenings to normalize participation in conservative communities. For instance, discussions around cancer screening in churches increased uptake among African migrants in Canada [125]. To translate these lessons into action, policymakers must fund community-designed interventions and prioritize universal access to culturally competent care [126]. By centering migrant voices in intervention design and addressing structural inequities while building trust in underserved communities progress can be made [127]. Somali refugee women in Minnesota had improved CCS after participating in mosque-based workshops co-designed with imams, which reframed pap smears as aligned with Islamic principles of health preservation, countering fatalism and modesty concerns [128]. In Australia, Korean migrant women increased biennial mammogram participation by using narrative videos in Korean, which normalized screenings as acts of self-care rather than cultural betrayal [50,129]. For example, bilingual community health workers who shared cultural backgrounds developed trust while explaining procedures in Spanish, dispelling myths that Pap smears threaten modesty or fertility for CCS among Latino migrants [90]. Arabic-speaking refugees in Canada engaged more with health services after receiving translated, faith-sensitive mobile app reminders co-developed with community leaders, which simplified instructions [130]. Effective strategies include co-designing interventions with cultural and community leaders to embed faith-based messaging and address fatalistic beliefs, providing bilingual community health workers to deliver home-based education [124],

leveraging familial trust to dispel myths, developing multimedia tools (e.g., culturally tailored videos, apps) for improved knowledge and health care [131,132].

Furthermore, prioritizing linguistically accessible and culturally competent communication can bridge understanding gaps, empower informed decision-making, and build trust in healthcare systems, particularly among non-English-speaking, refugee, and uninsured populations, where migration status and socioeconomic vulnerability intersect [133,134]. Multilingual education, including culturally adapted mobile apps, translated reminders, and telehealth pre-screening consultations, has reduced missed appointments and improved adherence among migrants. Targeting specific subgroups, like newer immigrants and male refugees, is crucial for developing effective interventions [8,135]. Effective interventions should target subgroups like newer immigrants and male refugees, using strategies such as simplified CRCS procedures, interpreter services, visual guides, and native-language workshops [136]. In Australia, using professional medical interpreters rather than family members during colonoscopy referrals for Arabic-speaking migrants enhanced trust and reduced stigma [137]. Other interventions include training healthcare providers to use professional interpreters, including gender-concordant ones for sensitive screenings, and developing multilingual digital tools (e.g., apps, videos) co-designed with migrant communities [138]. Additionally, leveraging trusted platforms to distribute low-literacy, visually guided materials (e.g., pictograms, flipcharts) and partnering with ethnic media outlets (e.g., radio, TV) to broadcast screening information in community languages is crucial for reaching populations with limited formal education [139].

Trust in healthcare providers and systems can overcome barriers to cancer screening among migrants by promoting culturally safe, transparent, and collaborative care. Additionally, innovative approaches, such as distributing home-based FIT kits through community clinics, have been effective in raising CRCS rates in the USA [140]. The influence of healthcare providers, along with care provision by female healthcare workers, has been shown to increase Pap test completion among South Asian migrants in the UK [38]. Similarly, East African migrants were more likely to engage with mammography after healthcare systems collaborated with imams to deliver faith-sensitive messaging, reassuring patients that screenings align with Islamic values of health preservation [141]. In Australia, Arabic-speaking migrants reported higher trust in colonoscopy referrals when clinics used professional interpreters reducing concerns about privacy [83]. These strategies respect cultural norms, reduce discrimination fears, and address systemic distrust through cultural competency training for providers, expanding access to interpreters and bilingual staff, and collaborating with trusted community figures to design faith-sensitive screening campaigns [142]. Other strategies include implementing continuity-of-care models where migrants see the same provider repeatedly to build rapport and developing confidential screening pathways to address privacy concerns [143,141]. Policy advocacy, such as state-funded programs that could offer free transportation and flexible hours, and has been effective in increasing mammography uptake among low-income migrants [144].By prioritizing trust-building through culturally congruent care, health systems can mitigate structural inequities and empower migrants to engage proactively with cancer screening.

Not having insurance among migrants significantly limits their access to cancer screening, especially in the USA, as many are unable to afford preventive care without coverage [145]. Lack of insurance limits migrants' access to cancer screenings, leading to delayed diagnoses and poor health outcomes, exacerbating health disparities and leaving them at higher risk of undiagnosed cancers. Addressing this requires expanding coverage, offering free screenings through community clinics, and integrating screenings into routine care, especially through primary care models and equity-focused programs that target uninsured migrants [146].

Family and community support for acculturation can address cancer screening barriers by fostering collaborative decision-making. Embedding screening support into social and cultural contexts can transform norms into catalysts for screening. Community-driven engagement, such as partnerships with ethnic organizations and trusted leaders, has been effective—e.g., Pacific Islander women in New Zealand increased CCS rates [147]. Social networks, like

family encouragement, have significantly influenced cancer screening, as shown by a refugee cohort in the USA where follow-through with colonoscopy referrals doubled [148]. Acculturation also plays a role, with each additional decade of residency associated with higher mammography uptake, highlighting the impact of adapting to host-country healthcare norms over time [149]. Collaboration through social and family networks is key; cross-sector partnerships, such as Australia's state-community campaigns linking screening to cultural festivals, have led to higher participation compared to clinic-based outreach alone [150]. Potential family and societal interventions for cancer screening could include intergenerational programs where family members act as health advocates, blending traditional values with support (e.g., youth health ambassadors guiding elderly relatives). Peer navigation networks, where acculturated migrants assist newcomers with clinic systems and address logistical concerns, could also be effective [151,152]. Leveraging cultural activities, such as festivals, for interactive health education—through storytelling and games—can help normalize cancer screenings [153]. Additionally, multilingual social media can effectively share survivor stories and practical guides, tailored to linguistic and cultural preferences [154].

Knowledge and awareness are crucial to enhancing risk perception of cancers and encouraging participation in screening programs among migrants. However, knowledge gaps persist, with newer immigrants demonstrating less understanding of CRCS guidelines compared to long-term residents [155]. Limited interpreter services and fragmented primary care access further exacerbate disparities, particularly in the US and UK, hindering equitable cancer screening participation among migrant populations [87,88]. Increasing awareness and knowledge about cancer screening among migrants in English-speaking countries is a crucial intervention to overcome barriers to screening [156,157]. Migrants often face cultural, language, and informational challenges that discourage them from seeking preventative healthcare, including cancer screenings. Providing culturally sensitive educational campaigns in multiple languages can help migrants understand the importance of regular screenings, while also addressing cancer risks specific to their communities [158]. For example, offering workshops, pamphlets, or online resources in migrants' native languages can bridge language gaps and provide relevant information [158,159]. Healthcare providers should be trained to understand cultural beliefs and fears that hinder screening participation, fostering trust. Addressing misconceptions or stigma through culturally tailored messaging and collaborating with community leaders can further enhance awareness and encourage screening [160]. Collaborating with community leaders and local organizations to raise awareness and promote screenings can strengthen trust and increase participation among migrant populations.

## Implications for policy and practice

Fig 3 summarizes the key suggestions for improving cancer screening among migrant populations in English-speaking HICs, under the following key themes [Fig 3]: Co-design interventions with migrant communities, train healthcare providers in cultural competence and the use of professional interpreters, provide accessible and affordable screening services, invest in multilingual and visual education materials, and leverage social and family networks to promote screening.

## Study limitations

This scoping review consolidated evidence from various studies on cancer screening among migrants in the USA, UK, Canada, and Australia, offering a broad understanding of screening uptake, implementation strategies, and their enablers and barriers. However, it lacked a pre-registered or publicly available protocol. Additional limitations include the absence of expert consultation to enhance the triangulation of findings. The review included both qualitative and quantitative studies, which may have contributed to heterogeneity in the results, limiting the possibility of sensitivity analyses or meta-analysis. Future studies should involve stakeholder engagement to strengthen the validity of the findings.

**Co-design interventions with migrant communities**

**Collaborate with community leaders**: co-develop educational content with faith and cultural leaders.
**Faith-based messaging**: normalize screenings through religious institutions (mosques, churches).
**Community focus groups**: involve migrants in focus groups to identify barriers.
**Cultural sensitivity in education**: tailor messages to community-specific values and norms.
**Train local community leaders**: empower trusted community members to promote screening.

**Training on cultural competence and professional interpreters**

**Cultural competency training**: healthcare workers with training on cultural sensitivities and gender issues.
**Use professional interpreters**: ensure interpreters instead of family members for confidentiality and accuracy.
**Gender-concordant care**: offer female providers for screenings, especially for conservative communities.
**Sensitivity to faith-based beliefs**: ensure providers understand and respect cultural and religious views.
**Confidentiality practices**: implement privacy-protecting measures in screenings to build trust.

**Accessible and affordable screening services**

**Free or low-cost screenings**: offer subsidized or free screening services, especially for uninsured migrants.
**Mobile screening units**: deploy mobile clinics to reach remote or underserved areas.
**Flexible hours**: provide screenings during evenings and weekends to accommodate working migrants.
**Transport assistance**: offer free or subsidized transport to screening sites.
**Primary care integration**: embed cancer screening into regular healthcare visits.

**Ensure multilingual and visual education materials**

**Multilingual campaigns**: provide health materials in multiple languages (e.g., Arabic, spanish).
**Visual aids**: use pictograms, infographics, and videos for easy understanding.
**Mobile apps**: develop apps with translated content, reminders, and booking options.
**Social media campaigns**: share culturally relevant posts and videos in community languages.
**Interactive platforms**: create quizzes, videos, and virtual guides tailored to migrant needs.

**Social and family networks to promote screening**

**Peer navigators**: train community members to guide and encourage others to get screened.
**Family health ambassadors**: involve family members, especially younger generations, in promoting screenings.
**Cultural health events**: organize health festivals, workshops, and fairs in collaboration with communities.
**Intergenerational programs**: use younger family members to educate older relatives about screenings.
**Social media engagement**: share survivor stories and screening info via multilingual social media platforms.

**Fig 3. Summary of interventions for cancer screening among migrants in English speaking high-income countries.**

## Conclusion

The uptake of cancer screening (breast, cervical, colorectal) varied and had low among migrants, refugees, and CALD populations in selected English-speacking high-income countries. There were several efforts in the implementation of strategies of cancer screening. Persistent barriers such as cultural stigma, language gaps, and mistrust in healthcare systems underscore the need for community-based, tailored interventions that enhance accessibility and trust. Addressing these challenges requires co-designed programs with community leaders, expanded access to culturally and gender-concordant providers, addressing communication gaps in the understanding of English by ensuring interpretation services, and sustained public health education to boost health literacy and encourage participation in screening services. Culturally sensitive and adaptive, equity focussed interventions on cancer screening should be prioritized by ensuring sustained funding, disaggregated data collection on the uptake of cancer screening and design and implementation of program on targeting diverse migrant groups.

## Supporting information

**S1 Table. PRISMA checklist.**
(DOCX)

**S2 Table. Search strategy (databases, search domains and search terms).**
(DOCX)

**S3 Table. Inclusion and exclusion criteria for selection of studies in the review.**
(DOCX)

**S4 Table. Data extracts from the included studies.**
(XLSX)

**S5 Table. Description of studies, design, population, country, migrant groups, and cancer types.**
(DOCX)

## Author contributions

**Conceptualization:** Resham B. Khatri, Yibeltal Assefa.

**Data curation:** Resham B. Khatri, Aklilu Endalamaw, Yibeltal Assefa.

**Formal analysis:** Resham B. Khatri, Yibeltal Assefa.

**Investigation:** Resham B. Khatri, Aklilu Endalamaw, Darsy Darssan.

**Methodology:** Resham B. Khatri, Aklilu Endalamaw, Darsy Darssan, Yibeltal Assefa.

**Project administration:** Yibeltal Assefa.

**Resources:** Resham B. Khatri, Yibeltal Assefa.

**Software:** Resham B. Khatri.

**Supervision:** Yibeltal Assefa.

**Validation:** Resham B. Khatri, Aklilu Endalamaw, Darsy Darssan, Yibeltal Assefa.

**Visualization:** Resham B. Khatri, Darsy Darssan, Yibeltal Assefa.

**Writing – original draft:** Resham B. Khatri.

**Writing – review & editing:** Resham B. Khatri, Aklilu Endalamaw, Darsy Darssan, Yibeltal Assefa.

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
