## [Decision Letter · Decision Letter 0]

26 Jun 2025

PONE-D-25-24566Cancer screening services among migrant populations: levels, implementation strategies, enablers, and barriersPLOS ONE

Dear Dr. Khatri,

Thank you for submitting your manuscript to PLOS ONE. After careful consideration, we feel that it has merit but does not fully meet PLOS ONE’s publication criteria as it currently stands. Therefore, we invite you to submit a revised version of the manuscript that addresses the points raised during the review process.

The manuscript is timely and adds further knowledge about access and use of cancer screening in migrant populations. Several weaknesses have been identified. First, there is a question of this study meeting all criteria for a systematic review. The authors are encouraged to revisit the study design and methodology and either make a stronger case for a systematic review or change the review approach. Second, themes from the literature review need better organization and further elaboration in the discussion (e.g., attention to specific concepts). Additional clarifications throughout the manuscript are also recommended. All comments from both reviewers should be carefully addressed and the manuscript revised accordingly.

We look forward to receiving your revised manuscript.

Kind regards,

Magdalena Szaflarski, PhD

Academic Editor

PLOS ONE

Journal Requirements:

2. During your revisions, please note that a simple title correction is required: Please include the term "Systematic review" in the title. Please ensure this is updated in the manuscript file and the online submission information.

3. We note that there is identifying data in the Supporting Information file <S5 Table.xlsx>. Due to the inclusion of these potentially identifying data, we have removed this file from your file inventory. Prior to sharing human research participant data, authors should consult with an ethics committee to ensure data are shared in accordance with participant consent and all applicable local laws.

-Location data

4. Please remove or anonymize all personal information (NAME), ensure that the data shared are in accordance with participant consent, and re-upload a fully anonymized data set. Please note that spreadsheet columns with personal information must be removed and not hidden as all hidden columns will appear in the published file.

Additional Editor Comments:

The manuscript is timely and adds further knowledge about access and use of cancer screening in migrant populations. Several weaknesses have been identified. First, there is a question of this study meeting all criteria for a systematic review. The authors are encouraged to revisit the study design and methodology and either make a stronger case for a systematic review or change the review approach. Second, themes from the literature review need better organization and further elaboration in the discussion (e.g., attention to specific concepts). Additional clarifications throughout the manuscript are also recommended. All comments from both reviewers should be carefully addressed and the manuscript revised accordingly.

Reviewers' comments:

Reviewer's Responses to Questions

**Comments to the Author**

1. Is the manuscript technically sound, and do the data support the conclusions?

Reviewer #1: Partly

Reviewer #2: Partly

2. Has the statistical analysis been performed appropriately and rigorously? 

Reviewer #1: No

Reviewer #2: Yes

3. Have the authors made all data underlying the findings in their manuscript fully available?

Reviewer #1: Yes

Reviewer #2: Yes

4. Is the manuscript presented in an intelligible fashion and written in standard English?

Reviewer #1: Yes

Reviewer #2: Yes

5. Review Comments to the Author

Reviewer #1: This manuscript addresses a timely and important topic—cancer screening disparities among migrant populations in high-income, English-speaking countries. It includes an impressive volume of literature and offers a comprehensive synthesis of implementation strategies, enablers, and barriers across diverse migrant groups and healthcare systems. However, there are several substantive issues that need to be addressed before this manuscript is suitable for publication:

1. I am disappointed that this review was not pre-registered with Prospero. It is difficult to say that the PRISMA checklist was followed for best practices when this is not the case.

2. While the manuscript refers to itself as a systematic review, the design, scope, and synthesis align more closely with a scoping review. The broad research questions, inclusion of diverse study designs, and use of thematic analysis are all characteristic of scoping reviews. I recommend the authors reclassify this as a scoping review and follow the PRISMA-ScR guidelines rather than PRISMA 2020.

3. Some of the five thematic domains appear to overlap. For instance, “culturally competent language communication” and “culturally tailored education” often describe similar interventions. The authors should consider consolidating overlapping themes and tightening the discussion

4. Although the authors applied the MMAT for study quality assessment, this information was not used in the synthesis or to qualify the conclusions. The authors should clarify whether study quality influenced the interpretation of findings. Alternatively, they could remove the MMAT altogether if not used to inform the analysis.

Reviewer #2: Notes from Reviewer to Authors

Cancer screening services among migrant populations: Levels, implementation strategies, enablers, and barriers

The authors bring attention to the importance of cancer screening and disparities among migrants settling in English-speaking, higher income countries. This important topic is relevant and timely. While important, the authors fall short in adequately explaining their methodology and discussing the barriers and enablers. Several actions will need to be taken before the paper is appropriate for publication.

Specific points have been noted under each section to address. Overall, while the review has an adequate number of articles included, the barriers are not adequately identified or justified as to why they are barriers. It would be appropriate to discuss why these barriers are important to be addressed and how they can be addressed rather than itemizing. What other enablers have been used in other interventions to address similar barriers? Some of your themes seemed to be very similar and could, perhaps, collapse into fewer ones. If this were done, the discussion could be stronger and more in-depth with fewer themes.

*I noted "partly" in question 1 because of the lack of sample size for most countries. Percentages reflected in Results and tables do not have any meaning without knowing the sample sizes.

Abstract

No issues

Introduction

• Some of the references establishing prevalence are old. References describing cancer prevalence should be more current, preferably in the past 3 years, as global prevalence data are published more frequently.

• Authors describe cancer prevalence globally yet focus on high income English-speaking countries. Prevalence in target countries should be noted.

Materials and Methods

• It is unclear at which point Covidence was introduced. Did Covidence screen and determine which articles to examine?

• Authors need to explain the decision to use Covidence or any machine learning (AI) tool, particularly because of the risk of bias. How was bias minimized in your review? In addition, Covidence was mentioned only once despite the program’s use in screening, data extraction, analysis, and synthesis. Did Covidence perform all these functions?

• Covidence should be included in the PRISMA flow diagram and the figure should have an appropriate title. Currently, it has no title.

Results

• Line 160 – what do you mean by “general migrants?”

• In looking for sample sizes and countries, I found the Excel sheet. It only shows the information about studies in Australia. What about the other 3 countries? You need to list these same data for the other countries. Table 4 shows percentages and not sample sizes, which are needed to interpret data.

Discussion

• Authors did not address the lack of access to health services and cost of screening for uninsured migrants and immigrants in the USA. Access issues are significant reasons for low uptake, later-stage diagnoses, higher mortality, etc. Further, many people with insurance cannot afford to access the insurance due to high copays and deductibles. This needs to be addressed.

• Cancer prevalence of LMICs from which migrants come could be compared to high-income countries in which they settle. This would give the paper depth.

• In lines 205 through 210 were unclear; I was left wanting to know about what sorts of stigma were experienced, which cultural taboos, etc.? How would community education programs specifically address these? These comments were begging for more discussion.

6. PLOS authors have the option to publish the peer review history of their article (what does this mean? ). If published, this will include your full peer review and any attached files.

**Do you want your identity to be public for this peer review?** For information about this choice, including consent withdrawal, please see our Privacy Policy .

Reviewer #1: **Yes: ** Taylor M. McCready

Reviewer #2: No

---

## [Author Response · Author response to Decision Letter 1]

20 Jul 2025

Response to the editor and reviewers

Response: Thank you for suggestions. We have formatted our manuscript as per the journal guidelines

2. During your revisions, please note that a simple title correction is required: Please include the term "Systematic review" in the title. Please ensure this is updated in the manuscript file and the online submission information.

Response: We thank editor for this advice. As suggested by the reviewers below to change the study design as scoping review, we have revised manuscript accordingly and included the Scoping review in the title.

3. We note that there is identifying data in the Supporting Information file <S5 Table.xlsx>. Due to the inclusion of these potentially identifying data, we have removed this file from your file inventory. Prior to sharing human research participant data, authors should consult with an ethics committee to ensure data are shared in accordance with participant consent and all applicable local laws.

-Location data

Response: Author team did not collect any primary data for this scoping review. The included excel sheet (S4 Table.xls) is prepared based on already published and publicly available individually de-identified data. All information included in the table are publicly available in the database and authors have only extracted from those studies for this review.

4. Please remove or anonymize all personal information (NAME), ensure that the data shared are in accordance with participant consent, and re-upload a fully anonymized data set. Please note that spreadsheet columns with personal information must be removed and not hidden as all hidden columns will appear in the published file.

Response: The spreadsheet does not contain any personal information or hidden columns, we have revised it accordingly. The names listed are the authors of the specific studies included in this review.

Additional Editor Comments: The manuscript is timely and adds further knowledge about access and use of cancer screening in migrant populations. Several weaknesses have been identified. First, there is a question of this study meeting all criteria for a systematic review. The authors are encouraged to revisit the study design and methodology and either make a stronger case for a systematic review or change the review approach. Second, themes from the literature review need better organization and further elaboration in the discussion (e.g., attention to specific concepts). Additional clarifications throughout the manuscript are also recommended. All comments from both reviewers should be carefully addressed and the manuscript revised accordingly.

Response: As per the reviewers’ suggestions, we have revised the design of this review as “Scoping review” and we have the methodology accordingly. Additionally, we have restructured and synthesised the findings and streamlined the discussion section accordingly. We thank you editor and both of the reviewers for their very constructive feedback on our manuscript.

Response to the reviewers

Reviewer #1: This manuscript addresses a timely and important topic—cancer screening disparities among migrant populations in high-income, English-speaking countries. It includes an impressive volume of literature and offers a comprehensive synthesis of implementation strategies, enablers, and barriers across diverse migrant groups and healthcare systems. However, there are several substantive issues that need to be addressed before this manuscript is suitable for publication:

Response: we thank reviewers for so constructive feedback for the revision.

1. I am disappointed that this review was not pre-registered with Prospero. It is difficult to say that the PRISMA checklist was followed for best practices when this is not the case.

Response: This is our limitation: we were unable to register this review in PROSPERO. However, we followed the PRISMA checklist for the review. We have taken your suggestions into account and, in accordance with the PRISMA-ScR guideline, revised the review as indicated in the feedback below.

2. While the manuscript refers to itself as a systematic review, the design, scope, and synthesis align more closely with a scoping review. The broad research questions, inclusion of diverse study designs, and use of thematic analysis are all characteristic of scoping reviews. I recommend the authors reclassify this as a scoping review and follow the PRISMA-ScR guidelines rather than PRISMA 2020.

Response: We included all types of studies in the review, i.e., qualitative, quantitative, and mixed methods studies. We fully agree with your assessment that this review aligns more closely with a scoping review. As a result, we have revised the review in accordance with the PRISMA-ScR checklist.

3. Some of the five thematic domains appear to overlap. For instance, “culturally competent language communication” and “culturally tailored education” often describe similar interventions. The authors should consider consolidating overlapping themes and tightening the discussion

Response: Thank you for your suggestion. We have revised the review based on your feedback, consolidating overlapping themes and streamlining the discussion.

4. Although the authors applied the MMAT for study quality assessment, this information was not used in the synthesis or to qualify the conclusions. The authors should clarify whether study quality influenced the interpretation of findings. Alternatively, they could remove the MMAT altogether if not used to inform the analysis.

Response: In accordance with the feedback provided in point 3, we have revised the study to align with a scoping review. As quality appraisal is not required within the broader framework of a scoping review, we have removed the MMAT tool that was previously used to appraise the included studies.

Reviewer #2: Notes from Reviewer to Authors

Cancer screening services among migrant populations: Levels, implementation strategies, enablers, and barriers

The authors bring attention to the importance of cancer screening and disparities among migrants settling in English-speaking, higher income countries. This important topic is relevant and timely. While important, the authors fall short in adequately explaining their methodology and discussing the barriers and enablers. Several actions will need to be taken before the paper is appropriate for publication.

Response: The authors' team thanks the reviewer for this constructive feedback. We have incorporated important feedback in the revision of our manuscript.

Specific points have been noted under each section to address. Overall, while the review has an adequate number of articles included, the barriers are not adequately identified or justified as to why they are barriers. It would be appropriate to discuss why these barriers are important to be addressed and how they can be addressed rather than itemizing. What other enablers have been used in other interventions to address similar barriers? Some of your themes seemed to be very similar and could, perhaps, collapse into fewer ones. If this were done, the discussion could be stronger and more in-depth with fewer themes.

Response: Thank you for the feedback. We have incorporated these insights into the revision, both in the results and discussion sections. The key enablers and barriers faced by migrants are summarized in the findings section under each broader implementation strategy. The findings are interpreted in the discussion section to draw key suggestions and interventions to address the barriers to cancer screening and explore how enablers can be customized to overcome these barriers. Finally, based on the discussion, the key interventions are summarized in a figure in the discussion section.

*I noted "partly" in question 1 because of the lack of sample size for most countries. Percentages reflected in Results and tables do not have any meaning without knowing the sample sizes.

Response: The details of the studies, sample size, and other characteristics of the study population included in this review are provided in the supplementary file (excel sheet).

Abstract: No issues

Response: thank you so much for the feedback

Introduction-• Some of the references establishing prevalence are old. References describing cancer prevalence should be more current, preferably in the past 3 years, as global prevalence data are published more frequently. Authors describe cancer prevalence globally yet focus on high income English-speaking countries. Prevalence in target countries should be noted.

Response: Recent data have been included in the introduction section. The prevalence of high-income, English-speaking countries is included, extracted from the World Health Organization's International Agency for Research on Cancer.

Materials and Methods: • It is unclear at which point Covidence was introduced. Did Covidence screen and determine which articles to examine? Authors need to explain the decision to use Covidence or any machine learning (AI) tool, particularly because of the risk of bias. How was bias minimized in your review?

Response: Covidence was used for the selection of titles and abstracts to identify the studies for full-text review. This has been clarified in the methods section. However, Covidence or any other AI tools were not used in the study selection process.

In addition, Covidence was mentioned only once despite the program’s use in screening, data extraction, analysis, and synthesis. Did Covidence perform all these functions? Covidence should be included in the PRISMA flow diagram, and the figure should have an appropriate title. Currently, it has no title.

Response: Selected records were imported into Covidence to screen titles and abstracts for identifying the list of studies. The first author (RBK) initially screened the titles and abstracts and identified the potentially eligible studies for full-text review. The second author (AE) independently evaluated the eligibility of each title and abstract of the included studies. RBK and AE then independently assessed the full text of relevant studies using an Excel sheet approved by the author team. The authors' team subsequently cross-checked for any discrepancies, discussed them, and reached an agreement through further review.

Results: • Line 160 – what do you mean by “general migrants?”

Response: General migrants, whose specific origin or country of birth was unspecified, were simply described as migrants.

• In looking for sample sizes and countries, I found the Excel sheet. It only shows the information about studies in Australia. What about the other 3 countries? You need to list these same data for the other countries. Table 4 shows percentages and not sample sizes, which are needed to interpret data.

Response: The sample size of the included studies was extracted and detailed in Table 4 (Excel sheet).

Discussion: • Authors did not address the lack of access to health services and cost of screening for uninsured migrants and immigrants in the USA. Access issues are significant reasons for low uptake, later-stage diagnoses, higher mortality, etc. Further, many people with insurance cannot afford to access the insurance due to high copays and deductibles. This needs to be addressed.

Response: The lack of insurance and access to health services, especially cancer screening, was discussed in the revision.

• Cancer prevalence of LMICs from which migrants come could be compared to high-income countries in which they settle. This would give the paper depth.

Response: In many LMICs, cancer screening programs are lacking, and the population has limited screening practices. In fact, people often have low awareness of the risks associated with cancer, which may prevent them from seeking cancer screening. If they migrate to high-income countries, they may still have low awareness and risk perception of these diseases. These factors could represent some inherent barriers to cancer screening practices among migrants from a demand-side perspective.

• In lines 205 through 210 were unclear; I was left wanting to know about what sorts of stigma were experienced, which cultural taboos, etc.? How would community education programs specifically address these? These comments were begging for more discussion.

Response: Stigma and cultural taboos related to cancer screening were discussed, with examples provided from migrant population such as Screening procedures such as pap smears and mammograms can be seen as violating cultural taboo, and concerns of privacy that discourage to seek screening of cancers.

---

## [Editor Report · Decision Letter 1]

23 Jul 2025

A scoping review on levels, implementation strategies, enablers, and barriers of cancer screening among migrants

PONE-D-25-24566R1

Dear Dr. Khatri,

We’re pleased to inform you that your manuscript has been judged scientifically suitable for publication and will be formally accepted for publication once it meets all outstanding technical requirements.

Kind regards,

Magdalena Szaflarski, PhD

Academic Editor

PLOS ONE
---

## [Editor Report · Acceptance letter]

PONE-D-25-24566R1

PLOS ONE

Dear Dr. Khatri,

I'm pleased to inform you that your manuscript has been deemed suitable for publication in PLOS ONE. Congratulations! Your manuscript is now being handed over to our production team.

Kind regards,

on behalf of

Dr. Magdalena Szaflarski

Academic Editor

PLOS ONE